# Long-Interval Intracortical Inhibition and the Cortical Silent Period in Youth

**DOI:** 10.3390/biomedicines11020409

**Published:** 2023-01-30

**Authors:** Kelly B. Ahern, Juan F. Garzon, Deniz Yuruk, Maria Saliba, Can Ozger, Jennifer L. Vande Voort, Paul E. Croarkin

**Affiliations:** 1Mayo Clinic Alix School of Medicine, 200 First St. SW, Rochester, MN 55905, USA; 2Mayo Clinic College of Medicine and Science, 200 First St. SW, Rochester, MN 55905, USA; 3Mayo Clinic Department of Psychiatry and Psychology, 200 First St. SW, Rochester, MN 55905, USA

**Keywords:** adolescent, cortical inhibition, GABA-B receptor, psychiatric disorders, transcranial magnetic stimulation

## Abstract

Background: The cortical silent period (CSP) and long-interval intracortical inhibition (LICI) are putative markers of γ-aminobutyric acid receptor type B (GABA_B_)-mediated inhibitory neurotransmission. We aimed to assess the association between LICI and CSP in youths. Methods: We analyzed data from three previous studies of youth who underwent CSP and LICI measurements with transcranial magnetic stimulation and electromyography. We assessed CSP and LICI association using Spearman rank correlation tests and multiple linear regression analyses adjusted for demographic and clinical covariates. Results: The sample included 16 healthy participants and 45 participants with depression. The general mean (SD) age was 15.5 (1.7), 14.3 (1.7) for healthy participants, and 15.9 (1.6) years for participants with depression. Measures were nonnormally distributed (Shapiro–Wilk, *p* < 0.001). CSP and LICI were not correlated at 100-millisecond (ρ = −0.2421, *p* = 0.06), 150-millisecond (ρ = −0.1612, *p* = 0.21), or 200-millisecond (ρ = −0.0507, *p* = 0.70) interstimulus intervals using Spearman rank correlation test. No correlations were found in the multiple regression analysis (*p* = 0.35). Conclusions: Although previous studies suggest that cortical silent period and long-interval intracortical inhibition measure GABA_B_ receptor-mediated activity, these biomarkers were not associated in our sample of youths. Future studies should focus on the specific physiologic and pharmacodynamic properties assessed by CSP and LICI in younger populations.

## 1. Introduction

Gamma-aminobutyric acid (GABA) is the predominant inhibitory neurotransmitter in the central nervous system. The ontogeny, functions, and neurophysiology of GABA signaling are incompletely understood in developing children and adolescents. Motor-evoked potential (MEP) studies with transcranial magnetic stimulation (TMS) and electromyography (EMG) can noninvasively measure cortical GABA function [1,2]. Metabotropic GABA type B (GABA_B_) receptors modulate action potentials via the regulation of potassium channels. Notably, GABA_B_ receptors regulate the activity of the serotonergic, noradrenergic, and dopaminergic neurotransmitter systems. The cortical silent period (CSP) and long-interval intracortical inhibition (LICI) have been used as indirect indices of GABA_B_ receptor-mediated inhibitory neurotransmission [2].

Dysfunctions in GABA neurotransmissions have been associated with mood disorders. Decreases in plasma, cerebrospinal fluid, and cortical GABA_B_ concentrations have been associated with major depressive disorder [3], which may lead to observable changes in CSP and LICI. Jeng et al. (2020) found that patients with treatment-resistant depression demonstrated significantly higher baseline estimated LICI scores (corresponding to more impaired LICI function) than patients with treatment-responsive depression and healthy controls, suggesting a possible association between depression treatment resistance and impaired LICI [4]. Although the literature appears to be divided, patients with depression have demonstrated changes to CSP duration (with both increased and decreased length) across several different studies [5].

Many previous studies of GABA_B_ receptor-mediated inhibition have focused on CSP and LICI methodology in general, but their findings were not consistent [6,7]. Much of that prior work was conducted with healthy adults. Less is known about the application and interpretation of CSP and LICI for studies in children and adolescents. Motor threshold is generally high in children but subsequently decreases during adolescence. Children may also have decreased cortical inhibition in general. Studies in adults suggest that LICI increases with age [8]. One prior study of children and adolescents also suggested that LICI increases in age, but only when depression is present [1,9]. However, studies of CSP and LICI in pediatric models of health and disease are limited.

While there is a small body of literature concerning the relationship between CSP and LICI in adults, to our knowledge, no rigorous pharmaco-TMS-EMG studies or large correlational studies of CSP and LICI in adolescents have been reported. Because of the complexities of neurodevelopment and shifts in GABA function during childhood, what CSP and LICI specifically measure in adolescents represents a knowledge gap. To address this gap in the literature, we sought to examine cross-sectional CSP and LICI measures in a sample of adolescents who were healthy or had depression to investigate the potential association between CSP and LICI in this population. On the basis of pharmacologic evidence linking both measures to cortical GABA_B_ receptor activity, we hypothesized that CSP and LICI would have an indirect association (as lower LICI and increased CSP both reflect increased cortical inhibition) adolescents with or without depressive disorders.

## 2. Materials and Methods

### 2.1. Study Design and Participants

This study was approved by the Mayo Clinic Institutional Review Board (Rochester, Minnesota) and the University of Texas Southwestern Medical Center Institutional Review Board (Dallas, Texas). We performed a secondary analysis of data collected from a pooled sample of participants from three previous studies [10,11,12] that used consistent methodology. The participants included adolescents (10–19 years) and young adults (<21 years). Healthy participants were recruited from the community, and those with depression were recruited from specialized pediatric mood clinics. A diagnosis of depressive disorder on the Kiddie Schedule for Affective Disorders and Schizophrenia—Present and Lifetime Version was requisite for inclusion in the depressed group. Before participating in any research activities, all participants younger than 18 years provided written informed assent, and a parent or guardian provided written informed consent. Participants younger than 18 years were also required to have a parent or guardian who was fluent in English. Participants 18 years or older provided written informed consent. Patients with a history of epilepsy, febrile or unprovoked seizures, intracranial tumors, intracranial surgical procedures, or any other contraindications specified by the TMS Adult Safety Screen [13] were excluded from the studies.

Participants underwent an assessment of TMS cortical activity measures within a single session. Each participant also completed a structured diagnostic interview with the Schedule for Affective Disorders and Schizophrenia for School-Age Children [14], and Children’s Depression Rating Scale-Revised (CDRS-R) scores were recorded, with higher scores corresponding to greater presence and severity of depressive symptoms [15].

### 2.2. TMS and EMG Procedures and Measurements

CSP and LICI measurements were collected in accordance with previously published methods [16,17]. All measurements were collected with identical methodology, equipment, and personnel supervised by one principal investigator at both sites (the neurophysiology lab relocated in 2011). Single- and paired-pulse TMS was applied to the left primary motor area to assess cortical inhibition activity. Participants were seated and wore earplugs during stimulation. MEPs of the right abductor pollicis brevis muscle were recorded by EMG electrodes attached to the skin surface over the muscle. All participants were allowed to continue taking any prescribed psychotropic medications, but stimulant use was not allowed on the day of TMS testing.

Two Magstim 200 stimulators connected by a BiStim module (The Magstim Company Ltd., Whitland, UK) were used for magnetic pulse generation. A figure-of-eight electromagnetic coil (with 70-mm diameter loops) held in a tangential position over the scalp was used to deliver pulses, and the cortical testing area for abductor pollicis brevis movement was located by using single pulses at constant intensities and shifting the coil until muscle movement was visually observed. The specific hotspot area was identified by recording the MEPs as the coil was shifted in rostral, caudal, and lateral directions between stimuli. The optimal spot was determined at the site with the largest MEPs. The resting motor threshold was defined by progressively increasing the stimulus intensity until the MEP amplitude observed on EMG was above 50 μV in 5 of 10 trials [18].

During CSP measurements, participants were asked to contract their right abductor pollicis brevis at 20% of their maximum contraction (determined by a hand-held dynamometer), and the left primary motor cortex was stimulated with a single pulse at 140% of their resting motor threshold. CSP was calculated by averaging the duration of the TMS pulse-interrupted voluntary motor activity (from when the pulse started to when the contraction resumed) in 10 trials.

For LICI measurements, 2 suprathreshold TMS pulses (suprathreshold conditioning and test stimulus) were delivered to the abductor pollicis brevis muscle during rest. Both pulses were calibrated to result in 1.0-mV peak-to-peak MEP amplitudes that were separated by interstimulus intervals (ISIs) of 100, 150, and 200 milliseconds. The amplitudes of the conditioned MEPs (MEPs produced as a result of the test stimulus) were averaged for 10 trials at each ISI. The amplitude of the conditioned MEP was expressed as the ratio to the mean unconditioned MEP amplitude.

### 2.3. Statistical Analysis

In accordance with our primary study objective, our sample size calculation considered a regression model with six predictors and 80% power to reject the null hypothesis. A large to medium (Cohen *f*^2^ = 0.25) effect size was determined with a 2-sided significance level (alpha) of 0.05 in a sample size of 61 participants. The considered effect size was based on previous studies suggesting a strong to moderate association between CSP and LICI [19,20].

The demographic and clinical characteristics of the participants were summarized as mean (SD) for continuous variables and frequency (%) for categorical variables. CSP duration was the predictor (independent) variable, and LICI measurements at each ISI were the dependent variables. Inferential statistical analyses were intended to evaluate the linkage between LICI measurements (conditioned/unconditioned MEP amplitude ratios) at 100-, 150-, and 200-millisecond ISIs with CSP duration in each participant.

After a Shapiro–Wilk test was applied to assess the normality of the data for each variable, the association between CSP and LICI was examined with Spearman rank correlation coefficients for each of the ISIs for the total sample, healthy control group, and depression group. A multiple linear regression model was performed to analyze the association between CSP and LICI while adjusting for patient age, sex, and CDRS-R scores. Secondary analyses were performed using Spearman rank correlation coefficients for each of the ISIs, excluding participants above 19 years old. Spearman rank correlation coefficient tests were used to evaluate associations between age with CSP and LICI at each ISI. Scatter plots were used to visualize CSP and LICI tendencies with age. All statistical procedures were 2-tailed and *p* < 0.05 was considered statistically significant. All statistical analyses were performed with BlueSky Statistics Software, v7.4 (BlueSky Statistics, Chicago, IL, USA).

## 3. Results

There were 77 individuals in the original pooled sample from which we drew our participants. Within that pooled sample, 16 participants were missing at least one CSP or LICI measure and were excluded from our analyses. The final sample consisted of 61 participants: 45 of whom had depression and 16 who were healthy (Table 1). Most participants were female (*n* = 34, 56%) and right-handed (*n* = 56, 92%). The mean (SD) age of the participants was 15.5 (1.7) years for the total sample, 14.3 (1.7) years for the healthy group, and 15.9 (1.6) years for the depression group, indicating that the sample was predominantly adolescent. The mean (SD) CDRS-R score for the total sample was 36.1 (15.5) at the time of the diagnostic interview.

Participants’ cortical inhibition measurements (CSP and LICI at 100 ms, 150 ms, and 200 ms ISIs) were nonnormally distributed. The mean CSP was 167 ms (40–300 ms range) for the total sample (Table 1 and Figure 1). The mean LICI was 0.5 at 100 ms ISI, 0.58 at 150 ms ISI, and 1.31 at 200 ms ISI (Table 1 and Figure 1). Age was not significantly correlated with any of the measures (Appendix A), but scatter plots exhibited tendencies of lower CSP and higher unconditioned/conditioned MEP ratios at 100 and 150 ms in higher ages (Appendix A).

CSP and LICI measurements at each ISI were not significantly associated with Spearman rank correlation tests for the total sample, healthy group, or depression groups (Table 2). A linear regression model adjusting for patient age, sex, and CDRS-R score as covariates indicated that LICI was not a predictive factor for CSP at any ISI (*R*^2^ = 0.1127, *p* = 0.35).

In secondary analyses excluding participants above 19 years old (*n* = 57), CSP and LICI at 100 ms ISI presented a significant negative correlation (rho = −0.285, *p =* 0.03) when assessed individually using Spearman rank correlation tests. There were no other significant correlations (Table 3).

## 4. Discussion

We investigated whether two TMS cortical inhibition measures that putatively index GABA_B_ receptor-mediated inhibitory neurotransmission (CSP and LICI) were correlated at 3 ISIs (100, 150, and 200 milliseconds) in a sample of healthy and depressed adolescents, and young adults. No significant associations were identified among individual tests of CSP and LICI at 3 different ISIs within the entire study sample. Furthermore, CSP and LICI were not significantly associated in a linear regression model that adjusted for patient age, sex, and CDRS-R score as covariates. Our secondary analyses revealed a weak negative correlation between CSP and LICI at 100 ms in adolescents when adults were removed from the analyses.

Studies of the association between CSP and LICI in adults have reported contradictory results and divergent conclusions, which may be due to differences in the protocols used in each study [19,20,21]. Tremblay et al. [21] reported that anodal transcranial direct current stimulation shortened CSP duration but did not affect LICI. In contrast, Benwell et al. [22] documented increased CSP durations and reduced LICI in healthy adults during muscle fatigue conditions. CSP and LICI had a moderately positive association in a study of athletes who had previous concussions, but intracortical inhibition also differed between athletes with previous concussions and healthy controls [19]. Therefore, pathophysiologic changes resulting from previous concussions may have altered the association between CSP and LICI.

Pharmacologic clinical trials [6,23] have reported that both CSP and LICI are potentiated by GABA_B_ agonists. Baclofen, a GABA_B_ agonist, and the nonspecific GABA enhancers tiagabine and vigabatrin potentiate both measures [6]. CSP and LICI were previously considered specific to GABA_B_ receptor-mediated cortical inhibition, but CSP is also apparently affected by GABA_A_ agonists [6]. In another study, lorazepam considerably reduced CSP in response to high-intensity test stimuli [23].

The GABAergic system has diverse and dynamic roles in the developing brain. In animal models, the GABAergic system undergoes substantial development until the end of adolescence [24]. GABA signaling is important for both excitatory and inhibitory neurotransmission in early life but transitions to primarily inhibitory neurotransmission by approximately 1 to 2 years of age in humans. This inhibitory neurotransmission gradually increases throughout childhood and adolescence and starts declining during adulthood [25,26,27]. These changes have been shown to be unrelated to the GABA levels in the cortex, and other contributory factors such as the changes in the GABAA receptor composition may be involved [27]. Additionally, GABAergic neurotransmission interacts with dopamine neurotransmission, which is also dynamic throughout childhood and adolescence [28]. Variations in GABA_A_ and GABA_B_ receptor expression levels, physiology, and function in young populations may explain the inconsistent associations between CSP and LICI in our study.

A previous report speculated that fast corticospinal fibers activated by TMS may rely on a different pathway than those implicated in voluntary tonic contraction, and dissimilar neural networks may trigger CSP or LICI [29]. The inhibitory components of the cortico-basal ganglia-thalamo-cortical loop are hypothesized to be the underlying mechanisms of the last segment of CSP. Zeugin and Ionta [29] suggested that abrupt and robust activation in the primary motor area during a conditioning stimulus stimulates the cortico-basal ganglia-thalamo-cortical loop reactions responsible for neuronal excitation inhibition after a TMS pulse. Globus pallidus and substantia nigra GABAergic projections to the thalamus suppress thalamocortical outputs. The recently described hyperdirect pathway (cortical–subthalamic–pallidal pathway) and the conventional indirect pathway of the basal ganglia are proposed to initiate this process [29]. When a voluntary movement is prepared, the hyperdirect pathway is activated ostensibly before the direct and indirect pathways [29]. The voluntary movement nature of CSP may be fundamental to the difference between CSP and LICI measures.

The last segment of CSP may mirror the centers of the activity upstream from the primary motor cortex, whereas LICI may reflect the action of the primary motor area [22]. The difference between our findings and those in previous CSP and LICI association studies in adults suggests a change in the proportion of the inhibitory components and different contributors to each measure in adolescents.

Our results support those of Tremblay et al. [21] and Benwell et al. [22], in which CSP and LICI represent distinct intracortical inhibitory elements. The single weak correlation observed between CSP and LICI after removing adults from the analysis may be explained by a time overlap between measures rather than sharing common intracortical inhibitory elements. The CSP is generally prolonged in the young [30], and our sample presented a similar tendency (Appendix A), with CSPs (mean = 167 ms) overlapping with the ISIs of LICI (100, and 150 ms). Cortical inhibitory states of individuals with prolonged CSP might be reflected in their LICI measures. However, the inconsistent correlations of these measures within our sample suggest an independent nature in their inhibitory mechanisms.

To our knowledge, this is the first study to examine the association between CSP and LICI in adolescents, although associations between these measures were reported in previous studies in adults [19,20]. A potential limitation of our study was the use of EMG rather than electroencephalography (EEG) for measuring CSP and LICI. Although Farzan et al. [20] reported an association between CSP and LICI with the use of TMS and EEG or EMG, the CSP–LICI association with EEG was stronger than that with EMG. Furthermore, our regression model provides limited interpretations with respect to MDD because motor-evoked potentials offer only a partial picture of the GABAergic system. The perigenual anterior cingulate cortex (PACC) and the dorsolateral prefrontal cortex (DLPFC) are the primary brain regions associated with GABAergic dysregulation in MDD [31], and unaltered motor-evoked potentials are not necessarily indicative of an absence of anomalies in those regions. Therefore, our use of EMG may have been a methodologic limitation. Our study may also be limited by the number of trials used to measure the CSP and LICI paradigms. While present recommendations suggest performing 8–10 trials for paired-pulse evoked potentials acquisition [18], acquiring a greater number of trials per paradigm might have reduced the variability of our sample. Moreover, our sample size estimation was based on adult studies [16,17], and the younger age of our population may influence this estimation. Our sample size might have been insufficient to identify any existing correlation, which is a limitation of our study.

## 5. Conclusions

Although current studies suggest that CSP and LICI both measure GABAergic receptor-mediated activity, these biomarkers were not associated in our study population of adolescents. Additional studies focusing on the specific physiologic and pharmacodynamic properties assessed by CSP and LICI in children and adolescents are needed.

## Figures and Tables

**Figure 1 biomedicines-11-00409-f001:**
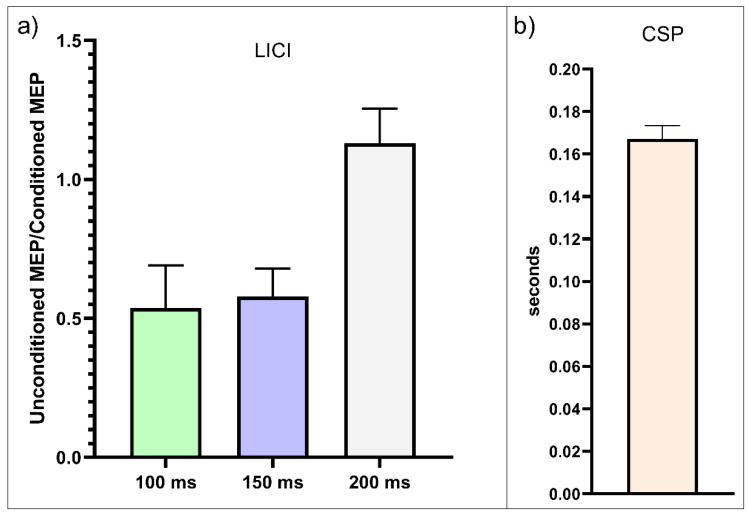
(**a**) Long-interval intracortical inhibition (LICI) by group. Mean conditioned/unconditioned MEP amplitude ratio values at interstimulus intervals (ISIs) of 100 ms, 150 ms, and 200 ms. Higher conditioned/unconditioned MEP amplitude ratio values in the LICI paradigm reflect reduced cortical inhibition. (**b**) Cortical silent period durations in seconds. Longer durations reflect greater inhibition.Error bars represent standard error estimates.

**Table 1 biomedicines-11-00409-t001:** Demographic and clinical characteristics.

Characteristic	Total Sample (*n* = 61)	Healthy (*n* = 16)	Depression (*n* = 45)
Age, *y*	15.5 (1.7)	14.3 (1.7)	15.9 (1.6)
Sex			
Female	34 (56)	5 (31)	29 (64)
Male	27 (44)	11 (69)	16 (36)
Right-handed	56 (92)	14 (88)	42 (93)
Race/ethnicity			
Asian	1 (2)	0 (0)	1 (2)
Black	7 (11)	5 (31)	2 (4)
Hispanic/Latino	2 (3)	1 (6)	1 (2)
Multiple/Other	6 (10)	2 (13)	4 (9)
White	45 (74)	8 (50)	37 (82)
CDRS-R score	36.1 (15.5)	19.6 (1.9)	41.9 (13.8)
CSP ^a^	167.2 (48.1)	179.8 (43.6)	162.7 (49.2)
LICI ^b^			
ISI = 100 ms	0.5 (1.2)	0.21 (0.3)	0.65 (1.37)
ISI = 150 ms	0.58 (0.79)	0.36 (0.51)	0.66 (0.86)
ISI = 200 ms	1.13 (1)	1.33 (1.2)	1.06 (0.88)
Duration of depression, *y*	NA	NA	2.0 (1.8)
Prescribed psychotropic medications	27 (44)	0 (0)	27 (60)
Family psychiatric history	44 (72)	0 (0)	44 (98)

Abbreviations: CDRS-R, Children’s Depression Rating Scale-Revised; NA, not applicable; CSP, Cortical Silent Period; LICI, Long-interval Intracortical Inhibition; ISI, Interstimulus Interval. Categorical data are summarized as No. (%) of participants, and continuous data (age, CDRS-R score, CSP, LICI, and duration of depression) are summarized as mean (SD). ^a^ CSP duration is expressed in milliseconds. ^b^ LICI conditioned motor evoked potential (MEP) amplitudes are expressed as ratios to the mean unconditioned MEP amplitudes.

**Table 2 biomedicines-11-00409-t002:** Association between the cortical silent period and long-interval intracortical inhibition in youth.

	Total Sample (*n* = 61)		Healthy (*n* = 16)		Depression (*n* = 45)	
Interstimulus Interval, ms	Spearman p	*p* Value	Spearman p	*p* Value	Spearman p	*p* Value
100	−0.2421	0.06	−0.4059	0.12	−0.1132	0.46
150	−0.1612	0.21	−0.318	0.24	−0.0042	0.98
200	−0.0507	0.70	−0.2971	0.30	−0.0004	>0.99

**Table 3 biomedicines-11-00409-t003:** Association between the cortical silent period and long-interval intracortical inhibition in adolescents.

	Total Sample (*n* = 57)		Healthy (*n* = 16)		Depression (*n* = 41)	
Interstimulus Interval, ms	Spearman p	*p* Value	Spearman p	*p* Value	Spearman p	*p* Value
100	−0.285	0.03 *	−0.4059	0.12	−0.172	0.28
150	−0.1944	0.14	−0.318	0.24	−0.0395	0.8
200	−0.076	0.05	−0.2971	0.3	−0.027	0.867

* Significant result with alpha set at 0.05.

## Data Availability

The data presented in this study are available on request from the corresponding author.

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
