# Peer review of "Long-Interval Intracortical Inhibition and the Cortical Silent Period in Youth"

_biomedicines, 2023, doi:10.3390/biomedicines11020409_

Round 1

Reviewer 1 Report

The current study was a secondary analysis of 3 prior studies that had used similar research methodology to measure the cortical silent period (CSP) and long-interval intracortical inhibition (LICI) in male and female adolescents. The overall purpose of the study was to examine the correlation between CSP and LICI and different interstimulus intervals (ISIs) as both of these measures have been reported to be associated with GABAB receptor activity in the primary motor cortex. The analyses were done on a total of 61 subjects with 16 of them being healthy adolescents and 45 having depression. Correlations were performed between CSP and LICI at 3 different ISIs. The strengths were most of the CSP and LICI methodology was appropriate and the writing was also pretty good with very few if any typos or grammatical errors.

While I am not sure based on the information provided in the study (some issues are unclear, see below) if there was a major fatal flaw, there were significant weaknesses that have to be addressed before the paper is publishable.

 Significant weaknesses include:

1.      While the over TMS methodology for measuring CSP and LICI were pretty solid, it is hard to understand why the authors chose to use the very subjective measure of the maximal thumb movement as the hotspot. This is rarely done unless it is a rTMS study and no EMG machine is present. Why not choose the location with the maximal MEP of the APB? The authors should give the reasoning for this. I don’t think this is a fatal flaw, but it really lowers my enthusiasm in the results.

2.      There needs to be figures added to the paper showing the raw average absolute values of the CSP and LICI. Why was this not done using simple bar graphs for instance? This should definitely be added so one can visualize the data (means and standard errors) and make sure the values correspond to those normally seen in the literature. Instead the authors only gave the range of CSPs for the whole sample, which is pretty puzzling. Just having the Spearman correlations in a table is not enough. Why not have correlation plots also so one can see the spread of the residuals? These should be added. Relatedly, what was the average % inhibition at each of the three ISIs for LICI?

3.      Age spread of participants and reporting. In the abstract, it says adolescents as the sample. In the methods it says children, adolescents, and young adults. I assume the methods section is more correct? Was there any attempt to see if CSP values differ across this wide age range (e.g. young adults vs children)? Afterall, several times in the paper the authors implied the CSP and LICI could change with age. Furthermore, the fact that the sample was supposedly adolescents was a major reason for doing the study. Yet, there appear to be some young adults and children involved. Did the authors not look at the possible age differences because the healthy sample only had 16 subjects? Why would only the overall average age of the whole sample be reported if it ranged from children to adolescents to young adults? Overall, it is very confusing how the age issue is reported. Was adolescence measured relative to puberty or just considered the age of 10-19? If the overall mean and SD reported in the paper for age is 15.7 plus or minus 1.7 and most of subjects were adolescent, why not just not use children or young adults? Surely it wasn’t too many participants from these two age groups?

4.      A few interrelated issues. I can’t see anywhere in the paper where there was some sort of physiological justification for quantifying CSP and LICI in participants with depression. Would this give any clinically relevant information? If so why was CSP and LICI not compared between the healthy and depression groups? Why were only correlations done within each group? It seems that one would want to do both and would do the basic group comparison of CSP and LICI before doing correlations within each group or the whole sample between CSP and LICI. Wouldn’t one want to know if there was a difference in group means if one were going to try to interpret any significant correlations or differences in correlations between CSP and LICI for either group?

5.      Do the authors think that taking the measures of CSP and LICI from primary motor cortex has any utility for depression considering the main brain area associated with depression is dorsolateral prefrontal cortex?

6.      Only using 10 trials for CSP and LICI is very questionable. Recent studies have suggested about 25 MEPs per group or condition is needed to get the true value. 10-12 trials were pretty much an absolute minimum in the early years of TMS and these low of values are rarely done nowadays. Once again, not a fatal flaw but very concerning. Any reason for such a low number? Doubling this number would have only taken 3-5 more minutes of experiment time most likely.

7.      Collectively, the above concerns make one wonder why the authors would report the data and do the analyses in the ways they did unless they were just trying to do a secondary analysis that was not well thought out just to get an additional publication. Many of the things above seem to be odd ways to separate, analyze, and report the data.

Reviewer 2 Report

The Manuscript: „Long-Interval Intracortical Inhibition and the Cortical Silent Period in Youth’’ by Kelly B. Ahern and colleagues reports on the analysis of previously documented data of youths who underwent CSP and LICI measurements with transcranial magnetic stimulation and electromyography. Based on the results of the study, the authors conclude that cortical silent period and long-interval intracortical inhibition measure GABAB receptor-mediated activity were not associated with cortical silent period in adolescents. The study is nicely performed and the results are appropriately interpreted. After going through the manuscript, I have a few comments for the authors:

1.     The number of participants is rather small (N, healthy: 16, depression: 45). I doubt whether the sample size was big enough to draw the statistical conclusions.

2.     How was the pooling of the samples of participants from previous studies performed. What was the selection criteria?

3.     Various previous studies have shown association of GABAergic receptor–mediated activity with CSP and LICI. This was, however, not reported in present study. Please briefly discuss the possible reason/s of this outcome in discussion section.

Round 2

Reviewer 1 Report

The authors have done a good job of answering all my previous comments and have made extensive changes to the manuscript. I think it is now ready for publication.